# AdapMTL: Adaptive Pruning Framework for Multitask Learning Model

## ABSTRACT

In the domain of multimedia and multimodal processing, the efficient handling of diverse data streams—such as images, video, and sensor data—is paramount. Model compression and multitask learning (MTL) are crucial in this field, offering the potential to address the resource-intensive demands of processing and interpreting multiple forms of media simultaneously. However, effectively compressing a multitask model presents significant challenges due to the complexities of balancing sparsity allocation and accuracy performance across multiple tasks. To tackle the challenges, we propose AdapMTL, an adaptive pruning framework for MTL models. AdapMTL leverages multiple learnable soft thresholds independently assigned to the shared backbone and the task-specific heads to capture the nuances in different components' sensitivity to pruning. During training, it co-optimizes the soft thresholds and MTL model weights to automatically determine the suitable sparsity level at each component in order to achieve both high task accuracy and high overall sparsity. It further incorporates an adaptive weighting mechanism that dynamically adjusts the importance of task-specific losses based on each task's robustness to pruning. We demonstrate the effectiveness of AdapMTL through comprehensive experiments on popular multitask datasets, namely NYU-v2 and Tiny-Taskonomy, with different architectures, showcasing superior performance compared to state-of-the-art pruning methods.

## CCS CONCEPTS

• **Computing methodologies** → **Image manipulation**; **Image processing**.

## KEYWORDS

pruning, multitask learning

## 1 INTRODUCTION

In the landscape of multimedia and multimodal processing [2, 40], Deep Neural Networks (DNNs) [46] have emerged as a pivotal technology, powering advancements across a spectrum of applications from image and video analysis to natural language understanding and beyond. Their profound ability to learn and abstract complex features from a range of media forms underpins their utility in diverse domains, including content categorization, recommendation systems, and interactive interfaces. However, as the complexity

**Unpublished working draft. Not for distribution.**

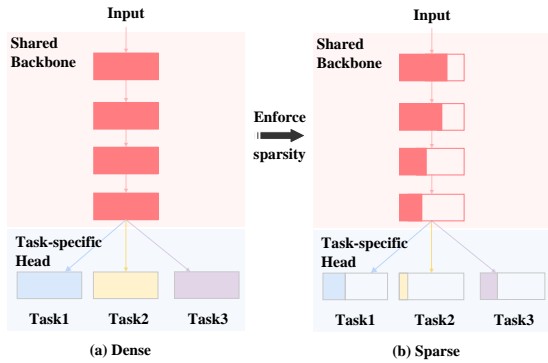

**Figure 1: Overview of pruning a dense multitask model. The red parts represent the shared backbone, and the leaf boxes represent the task-specific heads. In the sparse model, the blank spaces indicate the pruned parameters.**

of tasks grows, so does the demand for larger and more powerful models, which in turn require substantial computational resources, memory usage, and longer training times. This trade-off between performance and model complexity has led to a continuous pursuit of more efficient and compact CNN [24] architectures, as well as innovations in pruning techniques that can maintain high performance without compromising the benefits of the model's scale.

Pruning techniques [13, 19, 23, 25–27, 36, 48] have emerged as a promising approach to compress large models without significant loss of performance. These techniques aim to reduce the size of a model by eliminating redundant or less important parameters, such as neurons, connections, or even entire layers, depending on the method employed [9, 28, 61]. Parameter-efficient pruned models can provide significant inference time speedups by exploiting the sparsity pattern [14, 31, 56, 60]. These models are designed to have fewer parameters, which translates into reduced memory footprint and lower computational complexity (FLOPs) [31]. By leveraging specialized hardware and software solutions that can efficiently handle sparse matrix operations, such as sparse matrix-vector multiplication (SpMV), these models can achieve faster inference times [14, 39, 55]. Additionally, sparse models can benefit from better cache utilization, as they require less memory bandwidth, thereby reducing the overall latency of the computation [41, 60].

Although many techniques have been proposed in the past for pruning a single-task model, there are much fewer works in pruning a multitask model. Multitask models, which are designed to simultaneously handle multiple tasks, have become increasingly popular due to their ability to share representations and learn more effectively from diverse data sources [16, 64, 67]. These models have found wide-ranging applications where tasks are often related and can benefit from shared knowledge [65]. A compact multitask model, which is shown in Figure 1, has the potential to deliver

high performance across various tasks while minimizing resource requirements, making it well-suited for deployment on resource-constrained devices or in real-time scenarios.

Traditional pruning techniques, which are primarily focused on single-task models, may not be directly applicable or sufficient for multitask settings. Recent work has started to explore the intersection of multitask learning and pruning. Disparse [51] considered each task independently by disentangling the importance measurement and taking the unanimous decisions among all tasks when performing parameter pruning and selection. A parameter is removed if and only if it's shown to be not critical for any task. However, as the number of tasks increases, it becomes challenging to achieve unanimous selection agreement among all tasks, which could negatively affect the average performance across tasks. Thus, there is a need for novel compression approaches that are catered to the complexities of multitask models, taking into account the inter-dependencies between tasks, the sharing of representations, and the different sensitivity of task heads. Addressing these challenges is essential for advancing the development and deployment of efficient, compact multitask models.

To tackle the challenges, we conduct extensive experiments that reveal two valuable insights on designing an effective multitask model pruning strategy. *First, the shared backbone and the task-specific heads have different sensitivity to pruning and thus should be treated differently.* However, current state-of-the-art approaches do not adequately recognize this aspect, leading to equal treatment of each component during pruning, rather than accounting for their varying sensitivities. *Second, the change in training loss could serve as a useful guide for allocating sparsity among different components.* If the training loss of a specific task tends to be stable, we can prune more aggressively on that component, as the task head is robust to pruning. On the contrary, if the loss of a specific task fluctuates significantly, we should consider pruning less on that component since the training is less likely to converge at higher sparsity levels.

Motivated by these observations, we propose AdapMTL, an adaptive pruning framework for MTL models. AdapMTL dynamically adjusts sparsity across different components, such as the shared backbone and task-specific heads based on their sensitivity to pruning, while preserving accuracy for each task. This is achieved through a set of learnable soft thresholds [10, 23] that are independently assigned to different components and co-optimized with model weights to automatically determine the suitable sparsity level for each component during training. Specifically, we maintain a set of soft thresholds $\alpha = \{\alpha_B, \alpha_1, \alpha_2, ..., \alpha_T\}$ in each component, where $\alpha_B$ represents the threshold for the shared backbone and $\alpha_t$ represents the threshold for the $t$-th task-specific head. In the forward pass, only the weights larger than the threshold $\alpha$ will be counted in the model, while others are set to zero. In the backward pass, we automatically update all the component-wise thresholds $\alpha$, which will smoothly introduce sparsity. Additionally, AdapMTL employs an adaptive weighting mechanism that dynamically adjusts the importance of task-specific losses based on each task's robustness to pruning. AdapMTL does not require any pre-training or pre-pruned models and can be trained from scratch.

We conduct extensive experiments on two popular multitask datasets: NYU-v2 [47] and Tiny-Taskonomy [62], using different architectures such as Deeplab-ResNet34 and MobileNetV2. When compared with state-of-the-art pruning and MTL pruning methods, AdapMTL demonstrates superior performance in both the training and testing phases. It achieves lower training loss and better normalized evaluation scores on the test set across different sparsity levels. The contributions of this paper are summarized as follows:

(1) We conduct extensive experiments that reveal valuable insights in designing effective MTL model pruning strategies. These findings motivate the development of novel pruning strategies specifically tailored for multitask scenarios.

(2) We propose AdapMTL, an adaptive pruning framework for MTL models that dynamically adjusts sparsity levels across different components to achieve high sparsity and task accuracy. AdapMTL features component-wise learnable soft thresholds that automatically determine the suitable sparsity for each component during training and an adaptive weighting mechanism that dynamically adjusts task importance based on their sensitivity to pruning.

(3) We demonstrate the effectiveness of AdapMTL through extensive experiments on popular multitask datasets with different architectures, showcasing superior performance compared to state-of-the-art pruning and MTL pruning methods. Our method does not require any pre-training or pre-pruned models.

## 2 RELATED WORK

**Multitask Learning.** Multitask learning (MTL)[1, 4, 12, 64] aims to learn a single model to solve multiple tasks simultaneously by sharing information and computation among them, which is essential for practical deployment. Over the years, various MTL approaches have been proposed, including hard parameter sharing[3], soft parameter sharing [58], and task clustering [22]. In hard parameter sharing, a set of parameters in the backbone model are shared among tasks while in soft parameter sharing, each task has its own set of parameters, but the difference between the parameters of different tasks is regularized to encourage them to be similar. MTL has been successfully applied to a wide range of applications, such as natural language processing [8, 18, 29], computer vision [17, 30, 44, 57], and reinforcement learning [42, 53]. Subsequently, the integration of neural architecture search (NAS) with MTL has emerged as a promising direction. NAS for MTL, exemplified by works like MTL-NAS [15], Learning Sparse Sharing Architectures for Multiple Tasks [50], and Controllable Dynamic Multi-Task Architectures [43], focuses on discovering optimal architectures that can efficiently learn shared and task-specific features. These approaches, including Adashare [52] and AutoMTL [63], demonstrate the potential of dynamically adjusting architectures to the requirements of multiple tasks, optimizing both performance and computational efficiency.

**Pruning.** Pruning techniques have been widely studied to reduce the computational complexity of deep neural networks while maintaining their performance. Early works on pruning focused on unstructured weight pruning [20, 25], where unimportant weights were removed based on a given criterion, and the remaining weights were fine-tuned. There are different kinds of criterion metrics, such as magnitude-based [20, 27], gradient-based [36, 37], Hessian-based [21], connection sensitivity-based [26, 33, 48], and so on.

Other works explored structured pruning [56, 66], which removes entire filters or channels, leading to more efficient implementations on hardware platforms. Recently, the lottery ticket hypothesis [13] has attracted considerable attention, suggesting that dense, randomly-initialized neural networks contain subnetworks (winning tickets) that can be trained to achieve comparable accuracy with fewer parameters. This has led to follow-up works [13, 32, 38] that provide a better understanding of the properties and initialization of winning tickets. Single-Shot Network Pruning (SNIP) [26] is a data-driven method for pruning neural networks in a one-shot manner. By identifying an initial mask to guide parameter selection, it maintains a static network architecture during training. Some other work, like the layer-wise pruning method [23], inspiringly attempts to learn a layer-wise sparsity for individual layers rather than considering the network as a whole. This approach allows for fine-grained sparsity allocation across layers. To reduce the total time involved in pruning and training, pruning during training techniques [11, 35, 39] have been proposed to directly learn sparse networks without the need for an iterative pruning and finetuning process. These methods involve training networks with sparse connectivity from scratch, updating both the weights and the sparsity structure during the training process.

**Pruning for Multitask Learning.** Recently, attention has shifted to the intersection of MTL and pruning techniques. A compact multitask model has the potential to deliver high performance across various tasks while minimizing resource requirements, making it well-suited for deployment on resource-constrained devices or in real-time scenarios. For example, MTP [6] focuses on efficient semantic segmentation networks, demonstrating the potential of multitask pruning to enhance performance in specialized domains. Similarly, the work by Cheng et al.[7] introduces a novel approach to multi-task pruning through filter index sharing, optimizing model efficiency through a many-objective optimization framework. Additionally, Ye et al.[59] propose a global channel pruning method tailored for multitask CNNs, highlighting the importance of performance-aware approaches in maintaining accuracy while reducing model size. Disparse [51] proposes joint learning and pruning methods to achieve efficient multitask models. However, these methods often neglect the importance of the shared backbone, leading to equal treatment of each component during pruning, rather than accounting for their varying importance. Our work aims to address this limitation by adaptively allocating sparsity across the shared backbone and task-specific heads based on their importance and sensitivity.

## 3 METHODOLOGY

In this section, we first present the notations and the definition of the multitask model pruning We then introduce the proposed adaptive multitask model pruning framework in Section 3.2 and describe the adaptive weighting mechanism in Section 3.3.

## 3.1 Preliminary

We formulate multitask model pruning as an optimization problem. Given a dataset $\mathcal{D} = \{(x_i;\ y_i^1, y_i^2, ..., y_i^T),\ i \in [1, N]\}$, a set of T tasks $\mathcal{T} = \{t_1\ t_2\ ..., t_T\}$, and a desired sparsity level $s$ (i.e. the percentage of zero weights), the multitask model pruning aims to find

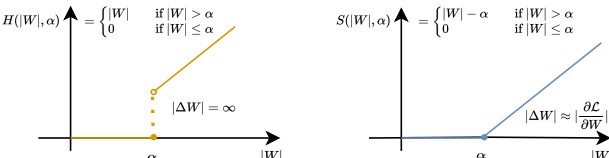

**Figure 2: Difference between hard and soft thresholding. Hard thresholding causes abrupt weight discontinuities during training, while soft thresholding ensures a smooth relationship for consistent learning.**

a sparse weight $W$ that minimizes the sum of task-specific losses. Mathematically, it is formulated as:

$$\min_{W} \mathcal{L}(W; \mathcal{D}) = \min_{W} \frac{1}{N} \sum_{i=1}^{N} \sum_{t=1}^{T} \mathcal{L}_t(f(W, x_i); y_i^t) \tag{1}$$
$$\text{s. t.} \quad W \in \mathbb{R}^d, \quad \|W\|_0 \leq (1-s) \cdot P,$$

where the $\mathcal{L}(\cdot)$ is the total loss function, $\mathcal{L}_t(\cdot)$ is the task-specific loss for each individual task $t$, $W$ are the parameters of neural network to be learned, $P$ is the total number of parameters and $\|\cdot\|_0$ denotes the $\ell_0$-norm, i.e. the number of non-zero weights. The key challenge here is how to enforce sparsity on weight $W$ while minimizing the loss. This involves finding an optimal balance between maintaining the performance of each task and pruning the model to achieve the desired sparsity level. We next describe our proposed adaptive pruning algorithm that can effectively handle the unique characteristics of multitask models and efficiently allocate sparsity across different components to preserve the overall model performance.

## 3.2 Adaptive Multitask Model Pruning

Multitask models typically have a backbone that is shared across tasks and task-specific heads. We observe that these different model components have different sensitivities to pruning and thus should be treated differently. The challenge lies in how to automatically capture the sensitivity of each model component to pruning and leverage the signal to automatically allocate sparsity across components. To address the challenge, we propose a component-wise pruning framework that assigns different learnable soft thresholds to each component to capture its sensitivity to pruning. The framework then co-optimizes the thresholds with model weights to automatically determine the suitable sparsity level for each component during training.

Specifically, we introduce a set of learnable soft thresholds $\alpha = \{\alpha_B, \alpha_1, \alpha_2, ..., \alpha_T\}$ for each component, where $\alpha_B$ represents the threshold for the shared backbone and $\alpha_t$ represents the threshold for the $t$-th task-specific head. The thresholds $\alpha$ are determined based on the significance and sensitivity of the respective components and are adaptively updated using gradient descent during the backpropagation process. The soft threshold $\alpha_t$ and sparse weight $W_t$ for each component can be computed as follows:

$$S(W_t, \alpha_t) = sign(W_t) \cdot ReLU(|W_t| - \alpha_t)$$
$$\alpha_t = sigmoid(\theta_{\text{init}}), \tag{2}$$

where $\theta_{\text{init}}$ is a learnable parameter that controls the initial pruning threshold $\alpha_t$. We will discuss the choice of $\theta_{\text{init}}$ in the supplementary material. The $ReLU(\cdot)$ function here is used to set zero weights. In other words, if some weights $|W_t|$ are less than the threshold $\alpha_t$, then the sparse version of this weight $S(w_t, \alpha_t)$ is set to 0. Otherwise, we obtain the soft-thresholding version of this weight.

The reason why we choose soft thresholding [54] rather than hard thresholding is illustrated in Figure 2. Soft parameter sharing is the best fit for our approach as it allows us to calculate the gradient and perform the backpropagation process more effectively.

AdapMTL reformulates the pruning problem in Equation 1 to find a set of optimal thresholds $\alpha = \{\alpha_B, \alpha_1, \alpha_2, ..., \alpha_T\}$ across different components as follows:

$$\min_{W, \alpha} \mathcal{L}(W, \alpha; \mathcal{D}) = \min_{W_t, \alpha_t} \frac{1}{N} \sum_{i=1}^{N} \sum_{t=1}^{T} \beta_t \cdot \mathcal{L}_t(f(S(W_t, \alpha_t), x_i); y_i^t)$$

$$\text{s. t.} \quad \alpha = sigmoid(\theta_{\text{init}}), \quad W \in \mathbb{R}^d, \quad \|W\|_0 \leq (1 - s) \cdot P, \tag{3}$$

where the $\beta_t$ represents the adaptive weighting factor for $t$-th task, which will be elaborated in Section 3.3.

We next describe how AdapMTL optimizes the problem in Equation 3. Considering a multitask model with T tasks, we divide the weight parameters into $W = \{W_B, W_1, W_2, ..., W_T\}$, where $W_B$ represents the weight parameters for the shared backbone and $W_t$ represents the weight parameters for the $t$-th task-specific head. We derive the gradient descent update equation at the $n$-th epoch for $W_t$ as follows:

$$W_t^{n+1} = W_t^n - \eta_n \frac{\partial \mathcal{L}(W, \alpha; \mathcal{D})}{\partial W_t^n}$$

$$= W_t^n - \eta_n \frac{\partial \mathcal{L}(W, \alpha; \mathcal{D})}{\partial S(W_t^n, \alpha_t^n)} \odot \frac{\partial S(W_t^n, \alpha_t^n)}{\partial W_t^n} \tag{4}$$

$$= W_t^n - \eta_n \frac{\partial \mathcal{L}(W, \alpha; \mathcal{D})}{\partial S(W_t^n, \alpha_t^n)} \odot \mathcal{B}_t^n,$$

where $\eta_n$ is the learning rate at the $n$-th epoch. We use the partial derivative to calculate the gradients. As mentioned earlier, different task heads may have varying sensitivities to pruning and, consequently, may require different levels of sparsity to achieve the best accuracy. By setting a set of learnable parameters for each component and treating them separately during the backpropagation process, our component-wise pruning framework can effectively account for these differences in sensitivity and adaptively adjust the sparsity allocation for each component.

Although $\frac{\partial S(W_t^n, \alpha_t^n)}{\partial W_t^n}$ is non-differentiable, we can approximate the gradients using the sub-gradient method. In this case, we introduce $\mathcal{B}t^n$, an indicator function that acts like a binary mask. The value of $\mathcal{B}t^n$ should be 0 if the sparse version of the weight $S(W_t^n, \alpha_t^n)$ is equal to 0. This indicator function facilitates the approximation of gradients and the update of the sparse weights and soft thresholds during the backpropagation process. Mathematically, the indicator function is:

$$\mathcal{B}_t^n = \begin{cases} 0, & \text{if } S(W_t^n, \alpha_t^n) = 0, \\ 1, & \text{otherwise.} \end{cases} \tag{5}$$

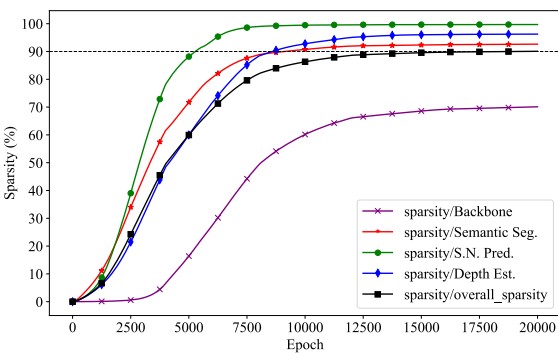

**Figure 3: Breakdown of component-wise sparsity allocation during training. We use the ResNet34 backbone and achieve 90% overall sparsity in the end.**

By updating the sparse weights $W_t$, and similarly the soft thresholds $\alpha_t$, for each component in this manner (the derivation process is provided in the supplementary material), the framework can effectively and discriminatively allocate sparsity across the multitask model. By taking into account the significance and sensitivity of each component, this approach ultimately leads to more efficient and accurate multitask learning.

### 3.3 Adaptive Weighting Mechanism

This subsection introduces the adaptive weighting mechanism that dynamically adjusts the weight of each task loss based on each task's robustness to pruning. The adaptive weighting mechanism determines the $\beta^t$ for the $t$-th task in Equation 3 during training.

The rationales behind the proposed adaptive weighting mechanism are two folds. First, if the training loss of a specific task $t$ tends to be stable, then we can assign a higher weighting factor $\beta_t$ and subsequently prune more aggressively on that component, as the task head is robust to pruning. On the contrary, if the loss of a specific task fluctuates significantly, we should consider pruning less on that component by lowering the weighting factor since the training is less likely to converge at higher sparsity levels. The weighting factor is learned in an adaptive way, eliminating the need for manual effort to elaborately fine-tune the hyper-parameters.

Second, the adaptive weighting mechanism should automatically consider different multitask model architectures as well. The ratio of backbone to task head weights, $\frac{W_{backbone}}{W_{head}}$, matters because it may be beneficial to focus more on pruning the task heads instead if the backbone is already highly compact. For example, in MobileNet-V2, the backbone has only 2.2M parameters, which is 25 times fewer than the task head.

We define a set of adaptive weights $\beta = \{\beta_B, \beta_1, \beta_2, ..., \beta_T\}$, where $\beta_B$ represents the weighting factor for the shared backbone, $\beta_t$ represents the weighting factor for the $t$-th task-specific head. The weighting factor can be formulated as follows:

$$\beta_t = \left( \frac{\Delta \mathcal{L}_t^{\text{window}} / \mathcal{L}_t}{\frac{1}{\text{T}} \sum_{t=1}^{\text{T}} (\Delta \mathcal{L}_t^{\text{window}} / \mathcal{L}_t)} \right)^{-1} \cdot \lambda \frac{|W_B|_{backbone}}{\sum_{t=1}^{\text{T}} |W_t|_{head}}. \tag{6}$$

Here, $\Delta\mathcal{L}_t^{\text{window}}$ is the average deviation of the loss within the sliding window for the $t$-th task, which is divided by $\mathcal{L}_t$ to normalize the scale. We then divide it by the sum of all tasks to normalize between different tasks. The $(\cdot)^{-1}$ is a multiplicative inverse. $\lambda$ is a scaling factor, and we will discuss the choice of $\lambda$ for different architectures in the supplementary material. $|W_B|_{backbone}$, $|W_t|_{head}$ represent the weight parameters of shared backbone and $t$-th task-specific head, separately. The right ratio in the equation reveals the importance of each component by considering their relative parameterizing contributions to the overall model structure. The weighting factor $\beta_t$ is used to guide the pruning for the task-specific head, depending on the stability of its loss and its contribution to the model.

To make the multitask pruning more robust, we incorporate a sliding window mechanism that tracks the past loss values to calculate the average $\Delta\mathcal{L}_{\text{window}}$ in Equation 6 instead of relying solely on the variance between two adjacent epochs. This approach provides a more stable and reliable estimation of the fluctuations in the task losses, as it accounts for a larger number of samples and reduces the impact of potential outliers or short-term variations.

## 4 EXPERIMENTS

In this section, we first present an overview of the experiment settings, including datasets, tasks, evaluation metrics, loss functions, baselines for comparison, and training details in Section 4.1. Subsequently, we provide comprehensive quantitative experimental results in Section 4.2, comparing our approach with other state-of-the-art methods to demonstrate the superiority of our proposed method. We also analyze the sensitivity of different components and computational cost in Section 4.3. Finally, we add ablation studies to verify the effectiveness of the proposed method in Section 4.4.

### 4.1 Experiment Settings

*4.1.1 Datasets and tasks.* We conduct the experiments on two popular multi-task datasets: NYU-v2 [47], and Tiny-Taskonomy [62]. The NYU-v2 dataset is composed of RGB-D indoor scene images and covers three tasks: 13-class semantic segmentation, depth estimation, and surface normal prediction. The training set consists of 795 images, while the testing set includes 654 images. For the Tiny-Taskonomy dataset, the experiments involve joint training on five tasks: Semantic Segmentation, Surface Normal Prediction, Depth Prediction, Keypoint Detection, and Edge Detection. The training set includes 1.6 million images, while the test set comprises 0.3 million images. The training set includes 1.6 million images from 25 different classes, while the test set comprises 0.3 million images across 5 classes.

*4.1.2 Evaluation Metrics and Loss Functions.* We adopt a range of evaluation metrics for different tasks, evaluating the model performance at different sparsity levels to provide a comprehensive view of the model's effectiveness and robustness across tasks. On the NYUv2 dataset, there are totally three tasks. For Semantic Segmentation, we employ the mean Intersection over Union (mIoU) and Pixel Accuracy (Pixel Acc) as our primary evaluation metrics and use cross-entropy to calculate the loss. Surface normal prediction uses the inverse of cosine similarity between the normalized prediction and ground truth, and is performed using mean and median

angle distances between the prediction and the ground truth. We also report the percentage of pixels whose prediction is within the angles of 11.25°, 22.5°, and 30° to the ground truth. Depth estimation utilizes the L1 loss, with the absolute and relative errors between the prediction and ground truth being calculated. Again, We also present the relative difference between the prediction and ground truth by calculating the percentage of $\delta = max(\frac{y_{pred}}{y_{gt}}, \frac{y_{gt}}{y_{pred}})$ within the thresholds of $1.25$, $1.25^2$, and $1.25^3$. On the Taskonomy dataset, there are two more tasks. In the context of both Keypoint and Edge Detection tasks, the mean absolute error compared to the provided ground-truth map serves as the main evaluation metric.

In multitask learning scenarios, tasks involve multiple evaluation metrics with values potentially at different scales. To address this, we compute a single relative performance metric following the common practice [34] [49].

$$\triangle_{T_i} = \frac{1}{|M|}\sum_{j=1}^{|M|}(-1)^{l_j} \cdot (M_{T_i,j} - M_{DM,j})/M_{DM,j} * 100\% \quad (7)$$

where $l_j = 1$ if a lower value shows better performance for the metric $M_j$ and 0 otherwise. $M_{T_i,j}$, $M_{DM,j}$ are the sparse and dense model value of metric $j$, respectively. The $\triangle_{T_i}$ is defined to compare results with their equivalent dense task values and the overall performance is obtained by averaging the relative performance across all tasks, denoted as $\triangle_T = \frac{1}{T}\sum_{i=1}^{T}\triangle_{T_i}$, This metric provides a unified measure of relative performance across tasks. Eventually, by employing these diverse evaluation metrics, we can effectively assess the performance of our method as well as the counterparts across various tasks and datasets.

*4.1.3 Baselines for Comparison.* We compare our work with LTH [13], IMP [19], SNIP [26], and DiSparse [51]. For LTH, we first train a dense model and subsequently prune it until the desired sparsity level is reached, yielding the winning tickets (sparse network structure). We then reset the model to its initial weights to start the sparse training process. For IMP, we remove the least important weights, determined by their magnitudes, iteratively. For SNIP and IMP, we directly use the official implementation provided by the authors from GitHub. For DiSparse, the latest multitask pruning work and first-of-its-kind, we utilize the official PyTorch implementation and configure the method to use the DiSparse dynamic mechanism, which is claimed as the best-performing approach in the paper. We also train a fully dense multitask model as our baseline, which will be used to calculate a single relative performance metric Norm. Score.

We use the same backbone model at the same sparsity level across all methods for a fair comparison. In our work, we define overall sparsity as the percentage of weights pruned from the entire MTL model, which includes both the shared backbone and task-specific heads. We utilize Deeplab-ResNet34 [5] and MobileNetV2 [45] as the backbone models, and the Atrous Spatial Pyramid Pooling (ASPP) architecture [5] as the task-specific head. Both of them are popular architectures for pixel-wise prediction tasks. We share a common backbone for all tasks while each task has an independent task-specific head branching out from the final layer of the backbone, which is widely used in multitasking scenarios.

**Table 1: Comparison with state-of-the-art pruning methods on the NYU-V2 dataset using the Deeplab-ResNet34 backbone. Each pruning method enforces a consistent overall sparsity of 90%, with the $\triangle_T$ indicating the normalized performance of all three tasks to the baseline dense model's performance. We also report the evaluation metrics for each task and the sparsity allocation for each component.**

| Model | $T_1$ : Semantic Seg. | | | $T_2$ : Surface Normal Prediction | | | | | | $T_3$ : Depth Estimation | | | | | | Sparsity % | | | | $\triangle_T\uparrow$ |
|---|---|---|---|---|---|---|---|---|---|---|---|---|---|---|---|---|---|---|---|---|
| | mIoU↑ | pixel Acc↑ | $\triangle_{T_1}\uparrow$ | Error↓ Mean | Median | Angle \theta, within↑ 11.25° | 22.5° | 30° | $\triangle_{T_2}\uparrow$ | Error↓ Abs. | Rel. | \triangle, within↑ 1.25 | 1.25^2 | 1.25^3 | $\triangle_{T_3}\uparrow$ | Back bone | S. S. head | S.N.P. head | D. E. head | |
| Dense Model (baseline) | 25.54 | 57.91 | 0.00 | 17.11 | 14.95 | 36.35 | 72.25 | 85.44 | 0.00 | 0.55 | 0.22 | 65.21 | 89.87 | 97.52 | 0.00 | | - | | | 0.00 |
| SNIP [26] | 24.09 | 55.32 | -10.15 | 16.94 | 14.93 | 36.17 | 72.39 | 86.98 | 2.63 | 0.61 | 0.23 | 60.61 | 87.88 | 96.77 | -25.49 | 85.46 | 90.24 | 92.28 | 91.17 | -11.00 |
| LTH [13] | 25.42 | 57.98 | -0.35 | **16.73** | 15.08 | 35.20 | 72.35 | **87.22** | 0.41 | 0.57 | 0.22 | 60.93 | 88.64 | 96.20 | -12.92 | 78.32 | 90.54 | 95.21 | 95.49 | -4.29 |
| IMP [19] | 25.68 | 57.86 | 0.46 | 16.86 | 15.18 | 35.53 | 71.96 | 86.26 | -1.77 | 0.56 | 0.22 | 65.23 | 89.29 | 97.53 | -3.82 | 74.98 | 92.34 | 97.23 | 95.15 | -1.71 |
| DiSparse [51] | 25.71 | 58.08 | 0.96 | 17.03 | 15.23 | 35.10 | 71.85 | 86.22 | -4.48 | 0.57 | 0.22 | 64.93 | 88.64 | 97.20 | -5.76 | 75.07 | 90.41 | 98.51 | 94.86 | -3.10 |
| AdapMTL w/o adaptive thresholds | 25.59 | 57.53 | -0.46 | 17.26 | 15.75 | 36.21 | 71.53 | 85.91 | -7.06 | 0.58 | 0.22 | 62.52 | 87.12 | 96.50 | -13.68 | 79.12 | 89.37 | 96.85 | 95.74 | -7.07 |
| AdapMTL (ours) | **26.28** | **58.29** | **3.55** | 16.92 | **14.91** | **36.36** | **72.97** | 86.29 | **3.41** | **0.55** | **0.22** | **65.39** | **89.93** | **97.58** | **0.38** | 71.74 | 93.18 | 99.26 | 96.22 | **2.45** |

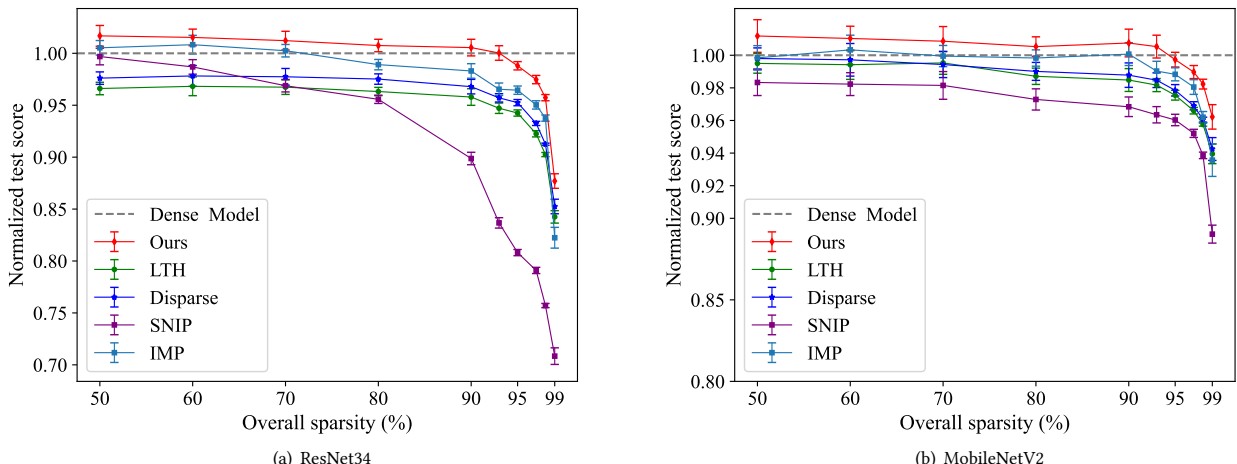

(a) ResNet34      (b) MobileNetV2

**Figure 4: Comparison of state-of-the-art methods, including DiSparse [51], LTH [13], SNIP [26], and IMP [19], on the NYUv2 dataset, evaluated with different MTL backbones and under various sparsity settings.**

## 4.2 Experiment Results

*4.2.1 Results on NYU-V2.* We first present the comparison results with state-of-the-art methods on the NYU-V2 dataset in table 1. Overall, AdapMTL outperforms all other methods by a significant margin across most metrics and achieves the highest $\triangle_T$. Recall that the major difference between our method and the baselines lies in our ability to adaptively learn the sparsity allocation across the components adaptively, maintaining a dense shared backbone (71.74%) while keeping the task-specific heads relatively sparse. Within the scope of our research, we characterize overall sparsity as the percentage of weights pruned from the entire MTL model, which includes both the shared backbone and task-specific heads.

SNIP [26] exhibits the lowest performance in the multi-task scenario because its pruning mask is determined from a single batch of data's gradient, which treats all components, including the shared backbone, equally. Since all input information passes through the shared backbone, accuracy loss in the shallow layers is inevitable, regardless of how well the task heads perform with relatively high

density. LTH's [13] winning tickets do not sufficiently focus on the backbone, as they intentionally create a dense surface normal prediction task head. Although this approach performs well on this specific task, the bias still causes an imbalance in the metrics across all tasks, resulting in a lower $\triangle_T$ score. IMP [19] achieves a good normalized score across all tasks. However, this method is trained in an iterative manner and prunes the model step-by-step, resulting in a significantly longer training time. DiSparse [51] learns an effective dense backbone by adopting a unanimous decision across all tasks. However, it falls short of differentiating the relative sensitivities between specific task heads, leading to an imbalanced normalized score among all tasks. Here, we add an additional row, AdapMTL without adaptive thresholds, to demonstrate the effectiveness of our approach. Rather than using multiple adaptive thresholds, this version utilizes a single shared threshold for all components. As expected, performance significantly deteriorates because a uniform threshold makes it hard to capture the nuances in different components' sensitivity.

**Table 2: Comparison with state-of-the-art pruning methods on the NYU-V2 dataset using the MobileNetV2 backbone. Each pruning method enforces a consistent overall sparsity of 90%, with the $\triangle_T$ indicating the normalized performance of all three tasks to the baseline dense model's performance. We also report the evaluation metrics for each task and the sparsity allocation for each component.**

| Model | $T_1$ : Semantic Seg. | | | $T_2$ : Surface Normal Prediction | | | | | | $T_3$ : Depth Estimation | | | | | | Sparsity % | | | | $\triangle_T\uparrow$ |
|---|---|---|---|---|---|---|---|---|---|---|---|---|---|---|---|---|---|---|---|---|
| | mIoU↑ | pixel Acc↑ | $\triangle_{T_1}\uparrow$ | Error↓ | | Angle \theta, within↑ | | | $\triangle_{T_2}\uparrow$ | Error↓ | | \triangle, within↑ | | | $\triangle_{T_3}\uparrow$ | Back bone | S.S. head | S.N.P. head | D.E. head | |
| | | | | Mean | Median | 11.25° | 22.5° | 30° | | Abs. | Rel. | 1.25 | 1.25^2 | 1.25^3 | | | | | | |
| Dense Model [5] (baseline) | 19.94 | 48.71 | 0.00 | 17.85 | 16.21 | 29.77 | 72.19 | 86.19 | 0.00 | 0.64 | 0.24 | 58.93 | 86.27 | 96.16 | 0.00 | - | | | | 0.00 |
| SNIP [26] | 18.96 | 46.93 | -8.57 | 18.33 | 16.97 | 28.93 | 71.21 | 85.78 | -12.03 | 0.64 | 0.25 | 56.75 | 85.71 | 95.33 | -9.38 | 78.46 | 88.19 | 92.08 | 90.25 | -9.99 |
| LTH [13] | 19.14 | 47.25 | -7.01 | 17.67 | 16.32 | 29.67 | 72.15 | 86.22 | -0.03 | 0.65 | 0.25 | 57.68 | 85.89 | 96.13 | -8.32 | 71.32 | 88.34 | 92.19 | 90.52 | -5.12 |
| IMP [13] | 18.76 | 48.12 | -7.13 | 18.71 | 16.68 | 29.63 | 71.76 | 85.91 | -9.11 | 0.64 | **0.23** | **59.75** | 86.52 | 96.31 | 6.00 | 68.49 | 88.07 | 95.13 | 87.74 | -3.41 |
| DiSparse [51] | 19.87 | 48.83 | -0.10 | 17.92 | 16.79 | 29.87 | 71.76 | 85.64 | -4.87 | 0.65 | 0.24 | 58.42 | 85.72 | 96.28 | -2.94 | 65.22 | 87.21 | 93.55 | 90.53 | -2.64 |
| AdapMTL w/o adaptive thresholds | 18.93 | 47.51 | -7.53 | 18.16 | 16.87 | 28.37 | 71.53 | 86.63 | -10.91 | 0.65 | 0.24 | 58.26 | 85.82 | 95.92 | -3.47 | 73.61 | 88.64 | 92.37 | 89.82 | -7.30 |
| AdapMTL (ours) | **20.16** | **49.14** | 1.99 | **17.53** | **15.96** | **30.16** | **72.36** | **86.51** | 5.25 | **0.64** | 0.24 | 59.03 | **86.57** | **96.38** | 0.75 | 52.74 | 86.18 | 94.72 | 90.76 | **2.66** |

Moreover, we extended our experiments to different model architectures to assess the model-agnostic nature of our method, using MobileNetV2 as an alternative architecture. The results, detailed in Table 2, show how AdapMTL adeptly manages the dense representation of MobileNetV2's compact backbone, ensuring it remains sufficiently dense (52.74 ) while enforcing higher sparsity in the task-specific heads. This is very important, especially with such backbone compact architectures where over-pruning the backbone can easily lead to significant degradation in accuracy. Our approach ensures that the backbone remains dense enough, thereby preserving overall performance.

*4.2.2 Results under various sparsity settings.* We show a comparison of results under different sparsity settings using different backbones, namely ResNet34 and MobileNetV2, as illustrated in Figure 4, where AdapMTL consistently demonstrates superiority over other methods. The normalized test score, following the common practice [34] [49], is obtained by averaging the relative performance across all tasks with respect to the dense model. We observe a slightly better performance for medium sparsity levels(from 50% to 80% ), which even surpasses dedicated dense multitask learning approaches despite the high sparsity enforced. This observation aligns with our assumptions and motivates the research community to further explore and develop sparse models. The score of SNIP drops significantly as higher sparsity levels (>90%) are enforced. This is because it fails to maintain the density of the shared backbone effectively.

*4.2.3 Results on Tiny-Taskonomy.* On the Tiny-Taskonomy dataset, which encompasses five distinct tasks, AdapMTL exhibits a more consistent performance across all tasks, as detailed in Table 3. Our method consistently achieved the highest scores in each task, unlike other methods which exhibited noticeable biases. The DiSparse method struggles to achieve unanimous decisions, particularly as the number of tasks increases, highlighting a key limitation in its approach.

The consistent superiority of AdapMTL across both NYUv2 and Tiny-Taskonomy datasets, and with different backbone architectures, highlights the effectiveness of our approach in achieving high sparsity with minimal performance degradation for multi-task models. More results on the other datasets, using the different architectures, can be found in the supplementary material.

**Table 3: Results on Tiny-Taskonomy dataset. T1: Semantic Segmentation, T2: Surface Normal Prediction, T3: Depth Prediction, T4: Keypoint Estimation, T5: Edge Estimation.**

| Model | $\triangle_{T_1}\uparrow$ | $\triangle_{T_2}\uparrow$ | $\triangle_{T_3}\uparrow$ | $\triangle_{T_4}\uparrow$ | $\triangle_{T_5}\uparrow$ | $\triangle_T\uparrow$ |
|---|---|---|---|---|---|---|
| SNIP | -11.2 | -15.7 | -9.4 | +1.2 | -2.8 | -7.58 |
| LTH | -9.9 | -1.3 | -10.7 | +0.5 | +3.1 | -3.66 |
| IMP | -6.3 | -9.7 | +3.1 | -1.1 | +2.4 | -2.32 |
| DiSparse | -1.6 | +1.2 | -3.9 | -1.5 | +4.2 | -0.32 |
| AdapMTL w/o adaptive thresholds | -8.7 | -12.6 | -4.7 | +0.2 | -1.4 | -5.44 |
| AdapMTL (ours) | **+2.8** | **+4.7** | **+1.5** | **+0.5** | **+4.9** | **+2.88** |

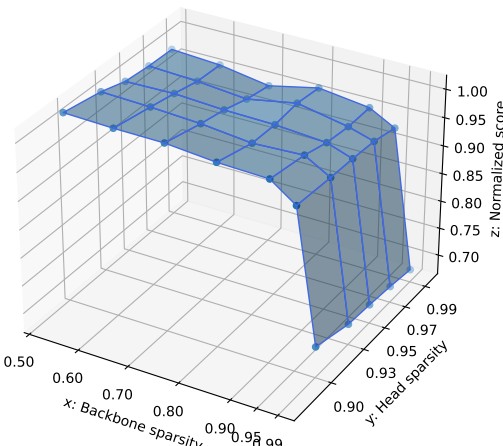

**Figure 5: Visualization comparing the sensitivity of the backbone and task head in a MobileNetV2 backbone MTL model. The y-axis represents the total sparsity of all task heads.**

## 4.3 Analysis

*4.3.1 Pruning sensitivity.* AdapMTL results in different sparsity for backbone parameters and task-specific parameters, indicating that it captures their different sensitivity to pruning. To compare the sensitivity to pruning between the shared backbone and task heads, we create a 3D plot, as shown in Figure 5. The x-axis represents the shared backbone sparsity from 50% to 99%, while the y-axis represents the total head sparsity for all three tasks from 90% to 99%. The z-axis represents the normalized score.

**Table 4: Computational cost of AdapMTL under various sparsity levels**

| Method | Sparsity (%) | Params | $\triangle_T \uparrow$ | FLOPs $\downarrow$ |
|---|---|---|---|---|
| Deeplab-ResNet34 | 0 | 197.6M | - | 56.32G |
| AdapMTL | 79.83 | 39.52M | 6.7 | 9.04G |
| AdapMTL | 85.01 | 29.64M | 4.3 | 7.84G |
| AdapMTL | 90.03 | 19.77M | 2.45 | 5.32G |
| MobileNetV2 | 0 | 155.2M | - | 37.32G |
| AdapMTL | 80.12 | 31.04M | 7.8 | 5.79G |
| AdapMTL | 85.03 | 23.28M | 5.2 | 4.21G |
| AdapMTL | 89.93 | 15.51M | 2.66 | 2.98G |

From the xz-plane, we can observe that the normalized score drops significantly when we prune the backbone at sparsity levels of 90% and higher. In contrast, from the yz-plane, we can see that the task heads are highly robust to pruning, as they maintain a good normalized score even when extreme sparsity levels are reached. This observation highlights the importance of preserving the shared backbone's density and suggests that pruning strategies should prioritize maintaining the backbone's performance while aggressively pruning the task-specific heads to achieve overall model sparsity.

*4.3.2 Computational cost.* The computational cost of the AdapMTL under varying sparsity levels is detailed in Table 4, which illustrates a significant reduction in both parameters and FLOPs as sparsity increases. These reductions highlight not only the adaptability of AdapMTL across different architectures but also its capability to maintain a balance between performance, measured by $\triangle_T$, and efficiency, evidenced by the substantial decrease in FLOPs. This balance is crucial for deploying high-performance models in resource-constrained environments. By leveraging specialized hardware and software solutions that can efficiently handle sparse matrix operations, such as sparse matrix-vector multiplication (SpMV), these models can achieve faster inference times [14, 39, 55].

## 4.4 Ablation Studies

We conducted ablation studies to validate the effectiveness of the proposed adaptive multitask model pruning (Section 3.2), and the adaptive weighting mechanism (Equation 6). We tested variations including models without adaptive thresholds, where all components share a single threshold, and models with only two adaptive thresholds, where the backbone has a unique threshold while other task heads share another. The results, presented in Table 5, highlight the critical role of adaptive thresholding. Models without adaptive thresholds showed significantly poorer performance, with a drastic decrease in $\triangle_T$, especially affecting tasks with higher sensitivity to pruning, such as Depth Prediction. Conversely, the full AdapMTL configuration, employing independent thresholds for each component, achieved the best $\triangle_T$ score. These variations help illustrate the impact and necessity of differentiated thresholding in multitask environments. The results confirm that our full AdapMTL setup, with all components active, performs superiorly across different settings, underscoring the indispensable nature of each proposed component.

**Table 5: Ablation Study on NYU-V2. T1: Semantic Segmentation, T2: Surface Normal Prediction, T3: Depth Prediction.**

| Model | $\triangle_{T_1} \uparrow$ | $\triangle_{T_2} \uparrow$ | $\triangle_{T_3} \uparrow$ | $\triangle_T \uparrow$ |
|---|---|---|---|---|
| w/o $\lambda$ (=5) | 1.26 | 1.74 | -1.83 | 0.39 |
| w/o sliding window | 3.07 | 2.84 | -0.49 | 1.81 |
| w/o adaptive thresholds | -0.46 | -7.06 | -13.68 | -7.07 |
| only 2 adaptive thresholds | -0.32 | -3.28 | -9.74 | -4.45 |
| AdapMTL | 3.55 | 3.41 | 0.38 | 2.45 |

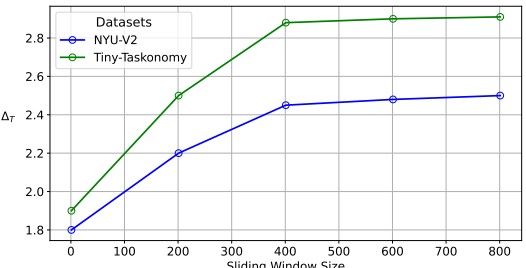

**Figure 6: Choice of sliding window size**

We have implemented a sliding window mechanism to enhance the robustness and accuracy of our pruning strategy. This mechanism is pivotal in tracking the loss values over a sequence of epochs to compute the average change in loss, $\Delta \mathcal{L}_{\text{window}}$, as formalized in Equation 6. By integrating this approach, we significantly mitigate the influence of abrupt variations and potential outliers that may occur in task-specific loss calculations. The sliding window, set at a size of 400 as demonstrated in Figure 6, represents an optimal balance between computational memory demands and the need for a comprehensive data scope. This size ensures that the model captures sufficient temporal loss information without excessive memory consumption, thereby maintaining efficiency.

## 5 CONCLUSION

In this paper, we propose a novel adaptive pruning method designed specifically for multitask learning (MTL) scenarios. Our approach effectively addresses the challenges of balancing overall sparsity and accuracy for all tasks in multitask models. AdapMTL introduces multiple learnable soft thresholds, each independently assigned to the shared backbone and task-specific heads to capture the nuances in different components' sensitivity to pruning. Our method co-optimizes the soft thresholds and model weights during training, enabling automatic determination of the ideal sparsity level for each component to achieve high task accuracy and overall sparsity. Furthermore, AdapMTL incorporates an adaptive weighting mechanism that dynamically adjusts the importance of task-specific losses based on each task's robustness to pruning. The effectiveness of AdapMTL has been extensively validated through comprehensive experiments on the NYU-v2 and Tiny-Taskonomy datasets with different architectures. The results demonstrate that our method outperforms state-of-the-art pruning methods, thereby establishing its suitability for efficient and effective multitask learning.

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
