# OpenReview forum: "AdapMTL: Adaptive Pruning Framework for Multitask Learning Model"
_acmmm.org/ACMMM/2024/Conference — MM2024 Poster_

### Official Review · Reviewer_qHFF · 2024-05-17

**Rating:** 3
**Confidence:** 3

**Summary:**

A pruning method designed for multi-task learning scenarios by introducing multiple learnable soft thresholds assigned to a shared backbone and specific task heads, by capturing the sensitivity of different components to pruning.

**Strengths:**

1.The introduction of learnable soft thresholds for both shared and task-specific components is innovative, enabling the model to adaptively prune each component based on its sensitivity. This allows for a more balanced and efficient pruning process.

**Limitations:**

1.Comparative Baseline Limitations:
The paper compares AdapMTL with several state-of-the-art methods but does not include comparisons with some newer pruning techniques or other advanced MTL optimization methods.Such as DynaShare: Task and Instance Conditioned Parameter Sharing for Multi-Task Learning.

2.There is very little in the article that deals with multimedia and multimodality.

3.In terms of article writing, does the article have a framework diagram that visually describes the entire algorithm.

4.In Fig. 2, the plot of soft thresholds is continuous, but with folds, which cannot be said to be smooth. Also is the meaning of the vertical coordinate here the number of weight parameters or what?

**Suitability:**

2

---

### Official Review · Reviewer_6vML · 2024-05-23

**Rating:** 4
**Confidence:** 2

**Summary:**

The paper introduces a method  AdapMTL to efficiently manage and compress multitask learning (MTL) models through adaptive pruning. The framework utilizes learnable soft thresholds which are independently set for the shared backbone and the task-specific heads of a multitask model. This allows for dynamic adjustment of sparsity levels during training. The method is validated with experiments on two datasets, demonstrating improved performance compared to existing SOTA pruning methods.

**Strengths:**

1. The approach is novel, including the adaptive weighting and the component-specific soft thresholds. AdapMTL addresses the unique challenges of pruning and keep balance in multitask scenarios

2. The framework is extensively tested on two datasets and two architectures with multiple ablation studies, providing a robust validation of its effectiveness.

**Limitations:**

1. The paper focuses solely on image-based tasks based on CNNs and does not explore or incorporate other modalities with different models. This unimodal approach may limit the generalizability and applicability of the proposed adaptive pruning framework to true multimedia or multimodal contexts.

2. The experimental results shows that even with the pruning, this method achieves better performance than the original dense models on almost all metrics and tasks. This is a little bit contrary to common sense and it’s not discussed in the paper.

**Suitability:**

2

---

### Official Review · Reviewer_SiSL · 2024-05-25

**Rating:** 4
**Confidence:** 2

**Summary:**

The author proposes AdapMTL, an adaptive pruning framework for MTL models that dynamically adjusts sparsity levels across different components to achieve high sparsity and task accuracy. Extensive experiments on popular multitask datasets with varied architectures demonstrate the effectiveness of AdapMTL.

**Strengths:**

(1) The paper is well-presented and easy to follow, featuring delicate images and well-formulated tables. The experiments demonstrate significant effectiveness with notable margins.

(2) The proposed methods make sense to me. The component-wise pruning framework, which assigns different learnable soft thresholds to each component to capture its sensitivity to pruning, is intriguing.

**Limitations:**

(1) The paper only focuses on unimodal computer vision multi-task scenarios, which are somehow not in line with the topic of ACM MM.

(2) The authors apply the proposed methods to relatively small models, including ResNet34 and MobileNet. I am curious whether these methods can scale to larger models, including LLMs.

(3) The paper lacks a deeper analysis of the motivations behind the proposed methods and their operational efficacy. I would appreciate further empirical or mathematical justification during the rebuttal.

**Suitability:**

2

---

### Meta-Review · Area_Chair_e6rn · 2024-07-04

**Recommendation:** Accept (Poster)
**Confidence:** 4

**Metareview:**

This paper initially received mixed ratings, with critiques primarily focused on its simple experiments limited to unimodal computer vision. However, during the rebuttal process, the authors included new experiments involving Large Language Models (LLMs). These additional experiments successfully addressed the reviewers' concerns, resulting in unanimously positive reviews. Therefore, the Area Chair recommends acceptance.